# Long-Distance Movement of *Solanum tuberosum* Translationally Controlled Tumor Protein (*StTCTP*) mRNA

**DOI:** 10.3390/plants12152839

**Published:** 2023-08-01

**Authors:** Brenda Beatriz Xoconostle-Morán, Beatriz Xoconostle-Cázares, Brenda Yazmín Vargas-Hernández, Leandro Alberto Núñez-Muñoz, Berenice Calderón-Pérez, Roberto Ruiz-Medrano

**Affiliations:** Departamento de Biotecnología y Bioingeniería, Centro de Investigación y de Estudios Avanzados, Instituto Politécnico Nacional, Avenida Instituto Politécnico Nacional 2508, Col. San Pedro Zacatenco, Ciudad de México 07360, Mexico; bbxoconostle@cinvestav.mx (B.B.X.-M.); bxoconos@cinvestav.mx (B.X.-C.); byvargas@cinvestav.mx (B.Y.V.-H.); leandro.nunez@cinvestav.mx (L.A.N.-M.); bcalderon@cinvestav.mx (B.C.-P.)

**Keywords:** TCTP, transient expression, *Solanum tuberosum*, tuber development

## Abstract

Long-distance signaling molecules in plants, including different RNA species, play a crucial role in the development and environmental responses. Among these mobile signals, the Translationally Controlled Tumor Protein (TCTP) mRNA is one of the most abundant. TCTP regulates cell-cycle progression and programmed cell death and is involved in responses to abiotic and biotic stress as well as plant regeneration, among other functions. Considering that the ability to induce plant regeneration is linked to a possible role of TCTP in vegetative propagation and asexual reproduction, we analyzed TCTP overexpression in a solanaceous plant model that can reproduce asexually by regeneration from stolons and tubers. Therefore, in this study, the effect of transient expression of *Solanum tuberosum* TCTP (*StTCTP*) on tuber development and vegetative propagation was described. *StTCTP* mRNA was shown to be transported long-distance. Additionally, transient overexpression of *StTCTP* resulted in sprouts with a greater diameter compared to control plants. Furthermore, the early stages of tuberization were induced compared to control plants, in which only mature tubers were observed. These results suggest a role of TCTP in vegetative propagation and asexual reproduction.

## 1. Introduction

Interorgan communication in plants is mediated by the long-distance transport of signaling molecules, including different RNA species. Indeed, intercellular transport of mRNA has been observed in cucurbits, Arabidopsis, sap-feeding insects like aphids, and between parasitic plants and their hosts [1,2,3,4,5,6,7], indicating its occurrence on a large scale across multiple species [2,6,8,9,10]. RNA molecules transported through the phloem translocation stream are involved in modulating leaf development, response to low phosphate availability, regulation of root architecture via auxin signaling, gene silencing, pathogen infection, and tuber induction [11,12,13,14,15,16,17,18,19,20]. Moreover, phloem-transported and graft-transmissible small interfering RNAs (siRNAs) mediate post-transcriptional and transcriptional gene silencing (PTGS and TGS, respectively) [17,21], whereas specific microRNAs (miRNAs) transported from shoots to roots control phosphate uptake as well as tuber formation [15,19]. Long noncoding RNAs (lncRNAs) are also likely signaling molecules since they are phloem transported in response to phosphate deficiency [22]. Collectively, this evidence supports the role of RNAs as long-range signaling molecules in plants, coordinating growth and responses to environmental cues [23].

Among the mobile mRNAs, mRNA encoding the Translationally Controlled Tumor Protein (TCTP or TPT1), also known as p21 and p23, is one of the most abundant and conspicuous [4,6,24,25]. TCTP is involved in a wide range of biological processes, and complementation between Drosophila and Arabidopsis mutants lacking *TCTP* genes provides evidence of conserved function in both taxa [26]. TCTP interacts with anti-apoptotic proteins to promote cell protection [16,27,28,29]; participates in cellular stress responses [30]; regulation of cell cycle in plants and animals [26,31]; contributes to developmental regulation [32,33]; influences protein synthesis and degradation, pathogen-induced cell death, photosynthesis, hormone signaling, response to drought stress, root development and architecture [34,35,36,37,38,39,40], and immune response in animals [41]. Additionally, TCTP displays non-cell autonomous activity in several biological systems [42].

Both TCTP protein and mRNA are known to undergo long-distance transport in cucurbits, Arabidopsis, and *N. tabacum* [6,25,43,44]. The presence of TCTP in the proteomes and transcriptomes of phloem exudates from other species suggests that this is a general phenomenon in vascular plants [8,9,45,46]. The m^5^C methylation of *AtTCTP1* mRNA is a key factor eliciting its long-distance transport over graft junctions [47]. The long-distance transport of the Arabidopsis *AtTCTP1* and *AtTCTP2* mRNA and protein is associated with adventitious root formation and lateral root formation, which potentially underlies the ability of this gene to induce whole plant regeneration [39,40,43,44]. Additionally, we have proposed the existence of two main types of TCTP in plants based on their predicted structure and the ability to induce in vitro plant regeneration [48]. AtTCTP2-like proteins, such as *Cucurbita maxima* TCTP (CmTCTP), share this in vitro plant regeneration activity, while AtTCTP1-like proteins lack this property [44]. It is possible that the ability to induce plant regeneration is linked to a possible role in asexual reproduction and vegetative propagation.

Several plant taxa, such as the *Solanum* genus, undergo asexual reproduction through specialized underground structures derived from the stem known as stolons, from which tubers develop in response to systemic signals produced in leaves which are in turn activated by environmental cues [49]. Tubers serve as storage organs from which sprouts develop, representing an intermediate stage in the vegetative propagation of potato [50]. Thus, members of this genus are excellent models for studying vegetative propagation and asexual reproduction in plants.

There are several shoot-derived signals involved in tuber formation in potato (*Solanum tuberosum*), such as FT-like proteins StSP6A and StSP3D, *StBEL5* mRNA (encoding a transcription factor which together with the POTH1 transcription factor activate *StSP6A* expression in leaves) [51], and miRNAs 172 and 156 [19,20,52,53,54], suggesting that tuber induction relies on a fail-safe mechanism triggered by redundant long-range signals [55].

The potato genome harbors a single *TCTP* gene (*StTCTP)*. The predicted structure of the encoded protein is similar to AtTCTP2-like proteins and thus may be involved in plant regeneration and asexual reproduction. Therefore, we analyzed whether agroinfiltration of a *StTCTP-GFP* expression construct in potato could cause phenotypic modifications related to tuber development or vegetative propagation. Our results showed that *StTCTP-GFP* mRNA is transported to apical leaves and roots, indicating its capacity to move over long distances, and the transient *StTCTP-GFP* expression in leaves induced changes in tuberization.

## 2. Results

### 2.1. GFP-Fused StTCTP Transcript Is Transported Long Distance

Previously, *AtTCTP2* was shown to be a functional gene and to be transported long distances through a graft union, in addition to promoting regeneration and increased biomass in tobacco [44]. Given their predicted structural similarity, we hypothesized that the *StTCTP* gene may play a role in regeneration and, thus, in vegetative reproduction in potato. To explore this possibility, the *StTCTP* open reading frame (ORF) fused to GFP was cloned under the control of the 35S promoter for agroinfiltration of leaves of 3-week-old potato plants. *Agrobacterium rhizogenes* was used because, in previous studies, we found that *CmTCTP* and *AtTCTP2* were able to induce whole plant regeneration in tobacco using this strain [25,44]. Subsequently, endpoint RT-PCR was performed to detect the transcript of the reporter gene in agroinfiltrated and apical (systemic) leaves. GFP was detected in the apical leaves of 24 out of 30 plants agroinfiltrated with the vector harboring the *35S::StTCTP-GFP* construct (Figure 1). In contrast, *GFP* mRNA was not detected in the apical leaves of plants agroinfiltrated with the vector harboring the *35S::GFP-GUS* construct (control plants) (Appendix A).

These results were corroborated by RT-qPCR of leaves from nine samples in which the *GFP* transcript was detected in apical leaves of agroinfiltrated plants with *35S::StTCTP-GFP* construct and compared with *35S::GFP-GUS*-harboring vector plant samples (Figure 2). *StTCTP* transcript was detected in agroinfiltrated tissues, apical leaves, and roots (Figure 2A). Additionally, the *35S::StTCTP-GFP* transcript was found in the apical leaves and roots of transformed plants with this construct (Figure 2B). In contrast, the *bar* mRNA was detected only in leaves of agroinfiltrated plants with either the control or the *35S::StTCTP-GFP* constructs (Figure 2C). Endogenous *StTCTP* mRNA was also detected in the apex, leaf, stem, root, and callus to determine the accumulation sites of this transcript (Appendix A). There was a greater accumulation of *StTCTP* mRNA in the stem and root and lower in the apex, leaf, and callus.

### 2.2. Phenotype of Plants Agroinfiltrated with 35S::StTCTP-GFP or 35S::GFP-GUS

The phenotype of the agroinfiltrated plants with *35S::StTCTP-GFP* or *35S::GFP-GUS* were evaluated to determine whether *StTCTP* generated changes in the tuberization stage and leaf area.

#### 2.2.1. *StTCTP* Induced an Increase in the Number of Tubers in Early Stages of Tuberization

To determine whether *35S::StTCTP-GFP* influence the phenotype of tubers, agroinfiltrated potato plants were grown and maintained until their formation. Biomass, diameter, and number of tubers per plant were measured (Appendix A). No statistically significant differences between plants agroinfiltrated with *35S::StTCTP-GFP* and control plants regarding tubers per plant, tuber diameter, or biomass were found. However, *StTCTP* appeared to increase tuber induction since we found eight plants treated with *35S::StTCTP-GFP* harbored tubers in immature stages of tuberization (79 immature tubers in total, stages I–VII [56]) compared to control plants, in which only mature tubers (stage VIII) were observed (Figure 3; Appendix A).

In addition, the formation of plantlets and a stolon in three out of thirty plants was observed after agroinfiltration with *35S::StTCTP-GFP* (in which the *GFP* amplicon was detected). In contrast, no plantlets or stolons were observed in plants agroinfiltrated with *35S::GFP-GUS* (0/30; Figure 4).

#### 2.2.2. *StTCTP* Promoted an Increase in Sprout Diameter

In order to analyze the phenotype of mature tubers of plants agroinfiltrated with *35S::StTCTP-GFP*, a sprout generation assay was carried out. First, all the sprouts of the *35S::StTCTP-GFP* plants showed purple pigmentation indicating strong anthocyanin accumulation, while those from plants agroinfiltrated with the control vector did not show any pigmentation (Figure 5A,B). Second, although sprout number per tuber and their length did not show a statistically significant difference, the sprout diameter was larger in the emerging tubers of *35S::StTCTP-GFP*-treated plants compared to those agroinfiltrated with *35S::GFP-GUS* (Figure 5C–E).

### 2.3. Molecular Docking Simulation Indicated Interaction between StTCTP and PTB1/6

Transport of phloem-mobile mRNAs such as *StBEL5* and *POTH1* is mediated by the polypyrimidine tract-binding proteins (PTB) 1 and 6, forming a complex with their 3′-untranslated regions [53,57,58,59]. Thus, it was of interest to determine if these proteins could also interact in silico with *StTCTP* mRNA. First, the *StTCTP*, *AtTCTP1*, *AtTCTP2*, *CmTCTP,* and *StBEL5* transcript 3D structures were predicted, then docking of these RNAs with PTB1 and PTB6 proteins was carried out in silico. The models with the best scores and criteria were chosen for further analysis (Appendix A). On the other hand, for 3D structure prediction of *StPTB1* and *StPTB6*, the models with the best scores and validation values were also selected (Appendix A). Finally, for the protein-RNA docking, the interaction models with the highest confidence scores were analyzed (Appendix A). The results suggest that *StTCTP* mRNA may, in fact, interact with these proteins, implying that it could be transported to roots, where it might potentially participate in the regulation of tuber formation and vegetative propagation. Hypothetical interactions of *AtTCTP1*, *AtTCTP2*, *CmTCTP*, and *StBEL5* (control) with PTB1 and PTB6 were also predicted (Appendix A).

## 3. Discussion

The long-distance transport of proteins and RNAs (e.g., miRNAs, mRNAs, siRNAs, and lncRNAs) appears to regulate processes that require communication between distant plant tissues. Examples of these processes include flower and tuber induction, which are mediated by homologs of the FT protein; tuber induction by miRNA172, miRNA156, and *StBEL5* mRNA; response to phosphate deficiency by miRNA399 and other mRNAs, as well as silencing by siRNAs; among many others [14,17,19,20]. It has been determined that TCTP protein and mRNA are transported long-distance and have been found in phloem proteomes and transcriptomes across different species. TCTP has the potential to induce regeneration and adventitious roots across graft junctions. However, it remains unclear whether TCTP function is linked to its mobility to distant tissues. *CmTCTP* and *AtTCTP2* have been shown to induce plant regeneration, suggesting their potential association with asexual reproduction in plants [56]. Considering the structural similarity between these proteins, it can be hypothesized that these share a common function [48].

Transient transformation assays indicate that the *StTCTP* mRNA functions in a non-cell autonomous manner. Indeed, our results demonstrate the bidirectional transport of *StTCTP* mRNA from agroinfiltrated leaves to apical leaves as well as roots. In our analysis, the ORF was used, as prior studies have established that *TCTP* mRNA transport does not require 5′- or 3′-UTR. In addition, it has been demonstrated that *AtTCTP1* ORF (spanning from the start codon to position 285) is able to mediate transcript mobility [43,47]. *GFP* mRNA levels in apical leaves of plants agroinfiltrated with *35S::StTCTP-GFP* were higher than in the *35S::GFP-GUS*-treated plants, indicating the presence of the recombinant transcript in the apical leaves of these plants (which was also observed by endpoint RT-PCR). It is important to indicate that the *StTCTP-GFP* fusion mRNA is around 1.2 kb, plus approximately 300 bp of the poly A tail, while the *GUS-GFP* fusion is around 2.5 kb (excluding the poly (A) tail), which could make the transport of this mRNA more difficult, although several studies have failed to detect entry of the *GUS-GFP* transcript into the phloem translocation stream [13,43]. This finding supports the notion that *StTCTP* mRNA is transported over long distances, as *CmTCTP* and *AtTCTP2* are.

Agroinfiltrated plants were maintained in the greenhouse until mature tuber emergence to determine the potential effects of *StTCTP* overexpression on their phenotype. In our study, no statistically significant differences were observed in tuber number per plant, biomass, or diameter. However, we observed an evident larger number of mature and immature tubers in the *35S::StTCTP-GFP*-treated plants (179 in total) compared to the control group (88 in total), supporting the notion that this gene is involved in the tuberization process. Importantly, *StTCTP*-treated plants harbored tubers at different stages of immature development, in contrast, to control plants, in which these were not observed. It must be emphasized that the number of mature tubers was similar (100 total mature tubers in *StTCTP-GFP* treated plants versus 88 in control plants), but no immature tubers were observed in the control plants. A similar trend towards an increase in aerial biomass of plants that were agroinfiltrated with *35S::StTCTP-GFP* was also observed, which is consistent with the observation that tomato *TCTP* overexpression in tobacco causes an increase in biomass [35]. The induction of tubers in early stages of development after agroinfiltration with the vector harboring *35S::StTCTP-GFP* suggested that *StTCTP* could play a role in tuber formation, although a scenario in which tuber development is inhibited by *StTCTP* is also possible, judging from the presence of immature tubers in the treated plants. Furthermore, we found leaf-to-root transport of *StTCTP* mRNA, presumably through the phloem, which is supported indirectly by the presence of *TCTP* mRNA and protein in phloem exudates and its expression in phloem in other species. Similarly, *StBEL5* mRNA is phloem mobile, enhancing tuberization by targeting genes that control growth [53]. It is becoming increasingly clear that there are multiple phloem-mobile signals that control tuberization, either its induction or development, although their hierarchy is less clear since it appears that there are several independent pathways regulating tuber development. Thus, *StTCTP* may be involved more directly in tuber development. It will be of interest to determine the pathway through which it acts in this process. It should be added that the *StTCTP* mRNA is likely translated in the roots, and the observed effects are caused by the protein rather than the mRNA.

*35S::StTCTP-GFP* mRNA levels are 1 to 2 orders of magnitude lower than the endogenous *StTCTP* mRNA (as determined by *GFP* mRNA accumulation) (Figure 2). While this is expected given the transient nature of the transformation method, it is interesting to note that even this modest increase of transcript was able to induce a phenotype in roots, i.e., induction of the immature development stages (I–VII) of tuberization. It is of great interest to analyze if both expression and long-distance transport of StTCTP protein and mRNA are tightly regulated. Another effect of *StTCTP* overexpression was the accumulation of anthocyanins, antioxidant compounds that promote tolerance to biotic and abiotic stress [60], possibly related to the TCTP function of tolerance to different types of stress in plants. Thus, it can be proposed that overexpression of *StTCTP* could lead to an abnormal phenotype due to the unregulated transport of mRNA. An important effect observed was the formation of plantlets and stolons, a common mechanism for asexual plant reproduction, which may bear relation to the ability of *CmTCTP*, *AtTCTP2,* and, possibly, *StTCTP* to induce whole plant regeneration. It must be mentioned that plant regeneration mediated by the former two requires *A. rhizogenes* and, more precisely, its *rol* genes. It will be of interest to determine if this phenomenon occurs independently of *A. rhizogenes* in a species that is able to reproduce asexually from stolons, such as potato.

The main abiotic factors that influence the tuberization process include short-day conditions and cold temperatures [61]. Interestingly, the Arabidopsis *AtTCTP1* gene is induced by cold conditions, while in *Pharbitis nil*, the transcript encoding a *TCTP* homolog was isolated from cotyledons under short-day conditions [62,63]. These findings highlight the potential of *TCTP* in regulating tuberization in response to environmental cues.

Although *StTCTP* is most actively expressed in stems and roots and at lower levels in leaves, it is not clear why it is transported to distant tissues. It could be interpreted that some extra input of the gene product from source tissues is required in apices. However, it is of great interest to determine source tissues for long-distance transport of *StTCTP* in wild-type plants.

Given that *StBEL5* mRNA interacts with StPTB1/6, which are members of the PTB family of RNA-binding proteins that regulate specific stages of development through interaction with phloem-mobile transcripts [58,59], the potential interaction of the *StTCTP* transcript with these proteins was analyzed through molecular docking. The results indicated that this transcript could bind to PTB1/6 according to in silico predictions, suggesting that *StTCTP* may also be transported and its activity regulated by the StPTB protein family. Furthermore, our results indicated that the phloem-mobile transcripts *AtTCTP1*, *AtTCTP2*, *CmTCTP,* and *StBEL5* are also predicted to bind PTB1/6.

The results described herein indicate that the *StTCTP* mRNA is transported from agroinfiltrated leaves to apical leaves and roots, where it elicits changes in tuber development, as well as generation of plantlets from roots, thus suggesting a role in vegetative propagation and, by extension, in asexual reproduction.

## 4. Materials and Methods

### 4.1. Plant Material

Potato (*Solanum tuberosum* L. cv. Cambray) tubers were disinfected with 1% Tween-20 and distilled sterile water, incubated in the dark until sprouts emerged (about 2 weeks), and planted in sterile soil (soil:peat moss:agrolite; 2:2:1) in pots 40 cm long and sheltered in greenhouse. The soil was stirred twice a week to allow aeration of emerging roots and tubers.

Callus induction was performed as described previously [64] for determination of endogenous *StTCTP* mRNA levels. Stem explants previously disinfected with 0.5% bleach and abundantly washed with water were placed in MS medium (1.0 MS salts, 2% sucrose, and 0.4% Gelrite Gelrite agar) supplemented with 0.5 mg/L 2,4-D and 2.0 mg/L BA and incubated in dark conditions until callus formation (10 days). All reagents used were purchased from Sigma-Aldrich (St. Louis, MO, USA) unless otherwise stated.

### 4.2. RNA Extraction and Vector Construction

RNA was extracted from 1 g of ground tissues (leaf, root, and callus) using the guanidine hydrochloride method [65]. First-strand cDNA was synthesized using Superscript III (Thermo Fisher, Waltham, MA, USA) following the manufacturer’s recommendations, and the *StTCTP* ORF was amplified using Takara ExTaq (Takara Bio USA, San Jose, CA, USA) with *StTCTP*-specific primers (without stop codon; Appendix A) the conditions of the endpoint PCR were as follows: 30 cycles of denaturation (98 °C, 10 s), primer annealing (60 °C, 30 s), and primer extension (72 °C, 1 min). The amplification product was cloned into the pCR8/GW/TOPO vector (Invitrogen, Thermo Fisher), and its orientation was confirmed by PCR, digestion, and sequencing. The entry clone was recombined into the pB7FWG2.0 binary vector through Gateway cloning to obtain the *StTCTP* ORF fused to the *GFP* ORF (*35S::StTCTP-GFP*) under the control of the CaMV 35S promoter. Similarly, the e35S promoter from pBUN4U6SM vector [66] was amplified by PCR which was introduced into the pBGWFS7 binary vector also by Gateway cloning for use as a control (*35S::GFP-GUS* vector). Both the *35S::StTCTP-GFP* and *35S::GFP-GUS* constructs were used to transform *A. rhizogenes* (strain K599) by electroporation, as previously described [53]. Candidate clones were verified by PCR.

### 4.3. Potato Transient Transformation

Transient transformation was carried out by agroinfiltration of 3-week-old potato after sprout emerged. Three leaves distal to apical and root meristems were chosen to perform the agroinfiltration procedure [67]. Briefly, *A. rhizogenes* K599 carrying the *35S::StTCTP-GFP* vector was cultured in YEB medium (1 g/L yeast extract, 5 g/L peptone, 5 g/L sucrose, 0.5 g/L MgCl_2_) supplemented with antibiotics (spectinomycin 100 mg/L) and incubated for 48 h at 28 °C at 150 rpm. Subsequently, the bacterial culture was pelleted by centrifugation at 3000× *g* for 15 min at 4 °C. The resulting pellet was resuspended in infiltration medium (10 mM MgCl_2_·7H_2_O, 10 mM 2-[N-morpholino] ethanesulfonic acid (MES) pH 5.6 and 200 μM acetosyringone) until reaching an OD600 equal to 0.8. *A. rhizogenes* suspension was injected into the abaxial surface of the leaf. One week post-infiltration, tissues were collected for detection by endpoint RT-PCR and RT-qPCR.

### 4.4. Endpoint RT-PCR

Plant tissues (agroinfiltrated leaves, apical leaves, and roots) were collected for RNA extraction as previously described. cDNA synthesis was carried out with WarmStart^®^ RTx Reverse Transcriptase (New England Biolabs; Beverly, MA, USA), according to the supplier’s recommendations. Briefly, the kit components (final concentrations: Isothermal Amplification Buffer (1X), dNTP Mix (0.5 mM), Random Primer Mix (6 µM), RNase Inhibitor-Murine (20 units), WarmStart RTx Reverse Transcriptase (0.25 µL) were homogenized with target cDNA at a concentration of 100 ng/µL. Mix was incubated for 5 min at 25 °C for annealing, 10 min at 55 °C for synthesis, and 10 min at 80 °C for enzyme inactivation. cDNA was used for endpoint PCR to detect the *GFP* and *GAPDH* transcripts (Appendix A) using Platinum^®^ Taq DNA Polymerase (Invitrogen, Thermo Fisher) following the manufacturer’s recommendations.

### 4.5. Quantitative RT-PCR (RT-qPCR)

*GFP*, *StTCTP*, *bar,* and *GAPDH* transcript levels were determined by RT-qPCR. Then, 50 ng/µL of RNA was mixed with the KAPA SYBR^®^ FAST qPCR Master Mix reagents following the user manual specifications (Merck; Rahway, NJ, USA). Specific primers for *GFP*, *StTCTP*, *bar* [68], and *GAPDH* (Appendix A) were used to set up the RT-qPCR reaction. The mix was incubated in Step One Plus Real-Time PCR System under the following conditions: the holding stage consisted of 5 min at 42 °C for reverse transcription; the next step was 5 min at 95 °C; the cycling stage comprised 95 °C per 5 s and 60 °C per 20 s with 40 cycles; the melt curve stage consisted of 15 s at 95 °C and 1 min at 60 °C; finally, 15 s al 95 °C to assure no additional products amplified in the reaction. Each sample was analyzed in triplicate. *GAPDH* was used to normalize transcript accumulation. The 2^−ΔΔCt^ method was used to calculate relative transcript accumulation [69].

### 4.6. Shoot Induction Assay

Tubers of plants agroinfiltrated with the *35S::StTCTP-GFP* and *35S::GFP-GUS* constructs were incubated at 4 °C in dark for two months. Subsequently, number of shoots per tuber, length and diameter of the emerging shoot, as well as size and weight of the tubers, tubers per plant were determined with the IC Measure program (https://www.theimagingsource.com/, accessed on 12 October 2022).

### 4.7. Prediction of Tertiary Structures

#### 4.7.1. Prediction of 3D Structures of mRNA

mRNA sequences of *StTCTP*, *AtTCTP1*, *AtTCTP2*, *CmTCTP*, and *StBEL*5 were obtained from the Phytozome (phytozome-next.jgi.doe.gov/, accessed on 18 November 2022) and NCBI databases (https://www.ncbi.nlm.nih.gov/, accessed on 19 November 2022) (Appendix A). Each sequence was uploaded to RNAFold web server (http://rna.tbi.univie.ac.at/cgi-bin/RNAWebSuite/RNAfold.cgi, accessed on 9 February 2023) selecting the minimum free energy (MFE) and partition function algorithm to obtain the secondary structure with its Vienna format [70]. The 3D structure of the mRNAs was generated in the 3dRNA web Server (http://biophy.hust.edu.cn/3dRNA, accessed on 16 February 2023) by entering the transcript sequence and the Vienna format, choosing the option to generate 5 optimized predictions [71,72], which were analyzed in MolProbity (http://molprobity.biochem.duke.edu/, accessed on 22 February 2023) with the Nucleic Acid criteria, Geometry, Chiral handedness swaps, and Tetrahedral geometry outliers to perform the comparisons [73]. In addition, EMDataResource Pseudotorsion Plot Beta Server (https://ptp.emdataresource.org/index.html, accessed on 25 February 2023) was used to evaluate the best models based on virtual torsion angles generating pseudotorsion plots for RNA that are analogous to Ramachandran plots for proteins [74]. The model with the best scores and criteria was chosen for further analysis.

#### 4.7.2. Prediction of 3D Structures of Proteins

Prediction of the hypothesized 3D structure of StPTB1 and StPTB6 [58,59] was performed in the AlphaFold Colab Notebook program (AlphaFold.ipynb), which is an artificial intelligence (AI) algorithm that starts from the primary sequence of a protein to predict the 3D protein structure by incorporating neural network architectures and training procedures based on the evolutionary, physical, and geometric constraints of protein structures [75,76]. The obtained model was evaluated and refined in the DeepRefiner web server (http://watson.cse.eng.auburn.edu/DeepRefiner/index.php, accessed on 3 February 2023), which uses deep learning to generate new models [77], which were analyzed using MolProbity scores for proteins and Saves v60 (https://saves.mbi.ucla.edu/, accessed on 3 March 2023), which is a server for programs commonly used in protein structure validation [78,79,80] selecting the models with the best scores (ERRAT, Verify 3D, PROCHECK and Ramachandran).

#### 4.7.3. Protein-RNA Interactions by Molecular Docking Simulation

Protein-RNA docking was performed in HDOCK (http://hdock.phys.hust.edu.cn/, accessed on 15 March 2023), based on a hybrid algorithm of template-based modeling and ab initio free docking [81,82]. PDB corresponding to the proteins PTB1 and PTB6 were introduced with the ligand in PDB of the *StTCTP*, *CmTCTP*, *AtTCTP1*, *AtTCTP2,* and *BEL5* mRNAs independently. The model with the highest confidence score was analyzed and visualized in UCSF Chimera and 1.16 and UCSF ChimeraX 1.5 [83,84].

### 4.8. Statistical Analysis

Phenotype quantification was analyzed using GraphPad Prism version 8.0.0 for Windows, GraphPad Software, San Diego, CA, USA (https://www.graphpad.com/, accessed on 27 February 2023) with the Mann–Whitney U test with a two-tailed *p*-value (*p* > 0.05).

## 5. Conclusions

TCTP has diverse functions related to cell proliferation, regulation of programmed cell death, response to pathogens, and regeneration in plants. In addition, some members of this family function in a non-cell autonomous manner, protein, mRNA, or both. In the present study, through the transient transformation of potato, we found evidence that the *StTCTP-GFP* transcript is transported long-distance from agroinfiltrated leaves to the apical leaves and roots. Furthermore, it may have a role in tuber development and in regeneration, which in turn, suggests a role in asexual reproduction in this plant species. Additional work is required to determine the pathway through which *StTCTP* may regulate these processes.

## Figures and Tables

**Figure 1 plants-12-02839-f001:**
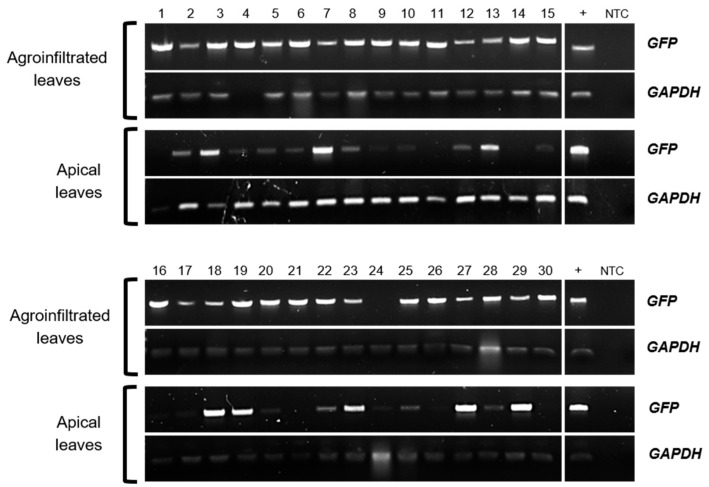
*GFP* and *GAPDH* mRNA detection in agroinfiltrated and apical leaves from potato plants transiently transformed with *35S::StTCTP-GFP* construct. Agarose gel electrophoresis of endpoint RT-PCR products from total RNA of transiently transformed plants (1–30). *GFP*: green fluorescent protein, reporter gene (720 bp). *GAPDH*: glyceraldehyde-3-phosphate dehydrogenase, endogenous gene (92 bp). +: positive control. NTC: non-template control.

**Figure 2 plants-12-02839-f002:**
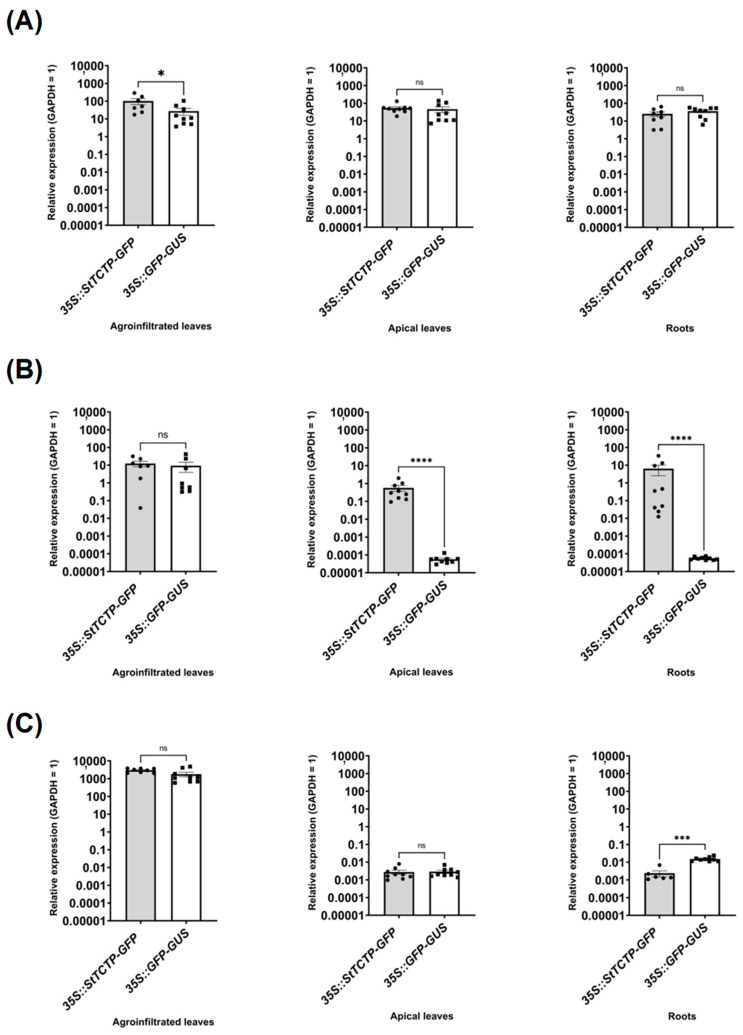
*StTCTP*, *GFP,* and *bar* mRNA relative accumulation levels in agroinfiltrated, apical leaves and roots from potato plants transiently transformed with *35S::StTCTP-GFP* or *35S::GFP-GUS*. Transcript levels of (**A**) endogenous plus *TCTP* derived from the *35S::StTCTP-GFP* construct, (**B**) *GFP* derived solely from the *35S::StTCTP-GFP* construct and (**C**) *bar* transcript derived from the selection marker gene present in the vector used for agroinfiltration with the *35S::StTCTP-GFP* or *35S::GFP-GUS* constructs. Data dispersion is represented with black circles and squares, respectively. n = 9 in all cases. Bars represent standard error of the mean. Asterisks symbolize significant differences according to Mann–Whitney tests: * *p* < 0.05, *** *p* < 0.005, and **** *p* < 0.0001; not significant (ns).

**Figure 3 plants-12-02839-f003:**
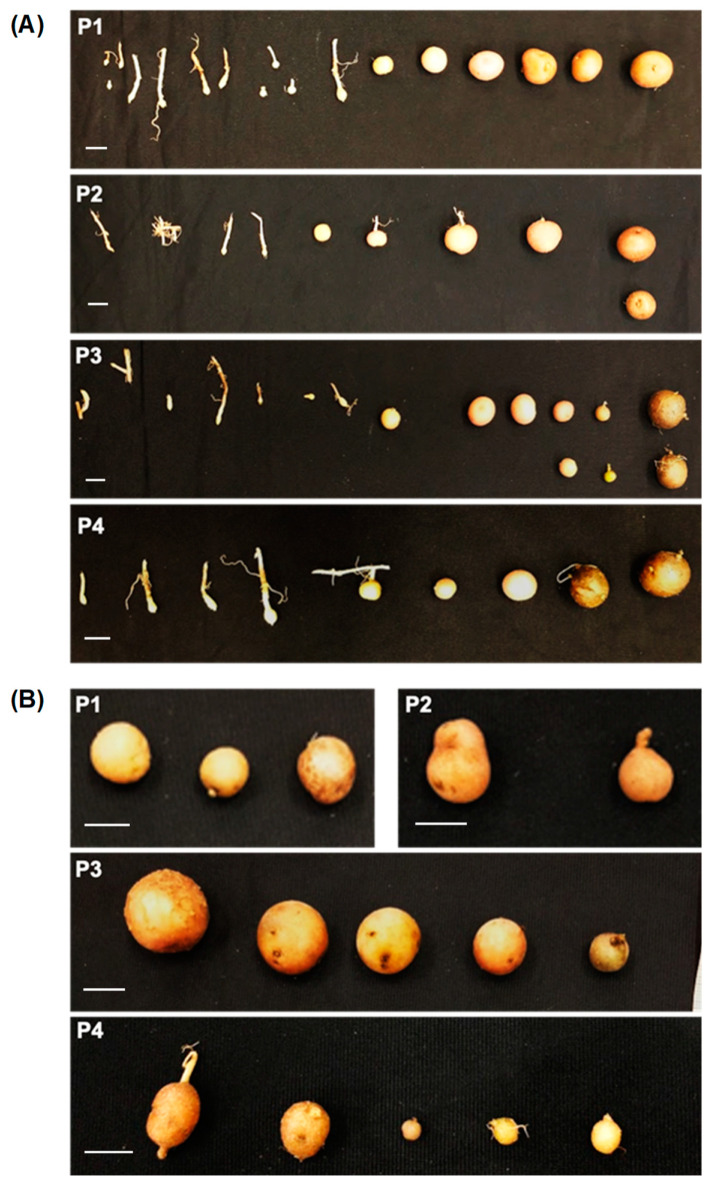
Induction of tuber formation in plants agroinfiltrated with *35S::StTCTP-GFP*. Tubers were collected two months after agroinfiltration. (**A**) Stolon stages up to mature tuber from four independent plants (P1–P4) agroinfiltrated with the *35S::StTCTP-GFP* construct. (**B**) Tubers that emerged from four plants (P1–P4) agroinfiltrated with *35S::GFP-GUS*, all of which were mature tubers; additionally, no stolons were observed. Scale bars = 1 cm.

**Figure 4 plants-12-02839-f004:**
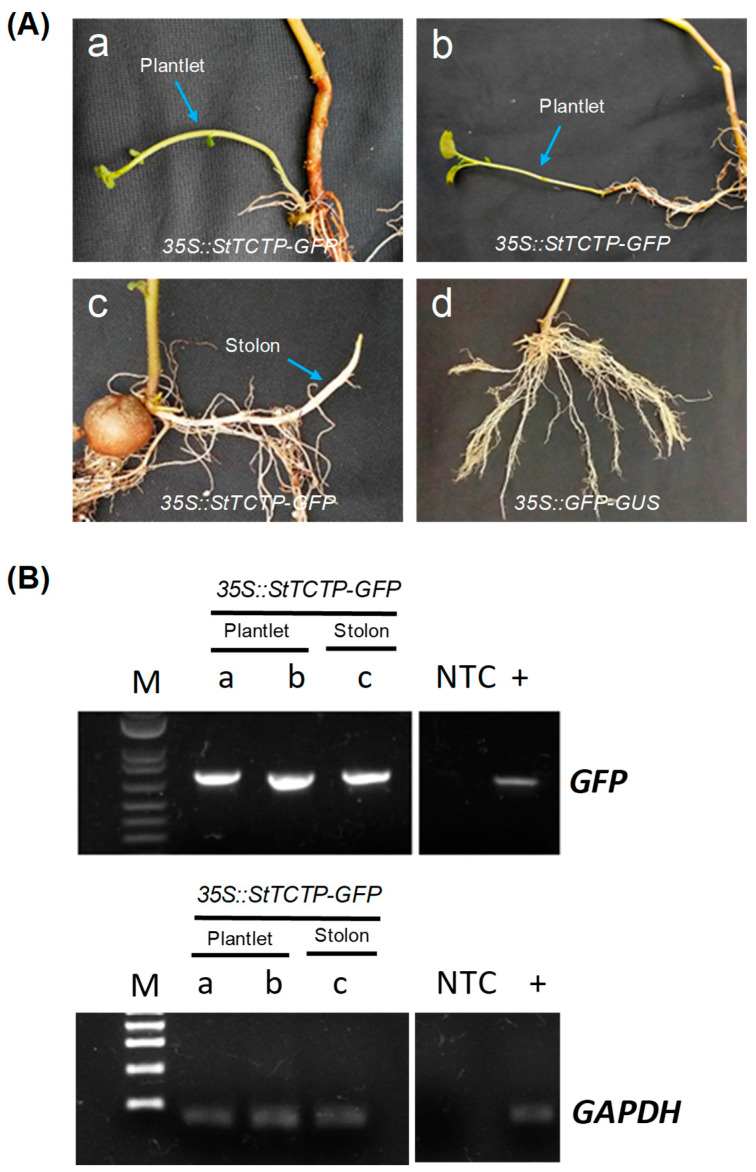
Induction of sprout formation as a result of agroinfiltration with *35S::StTCTP-GFP*. (**A**) Emerging plantlets (**a**,**b**) and stolon (**c**) two weeks post-agroinfiltration with *35S::StTCTP-GFP*. Plants agroinfiltrated with the *35S::GFP-GUS* did not show plantlet nor stolon formation two weeks after treatment (**d**). (**B**) Detection of *GFP* and *GAPDH* by endpoint RT-PCR of plantlets (a,b) and stolon (c). M: molecular marker, 1 kb Plus DNA ladder (Invitrogen, Waltham, MA, USA); +: positive control; NTC: non-template control.

**Figure 5 plants-12-02839-f005:**
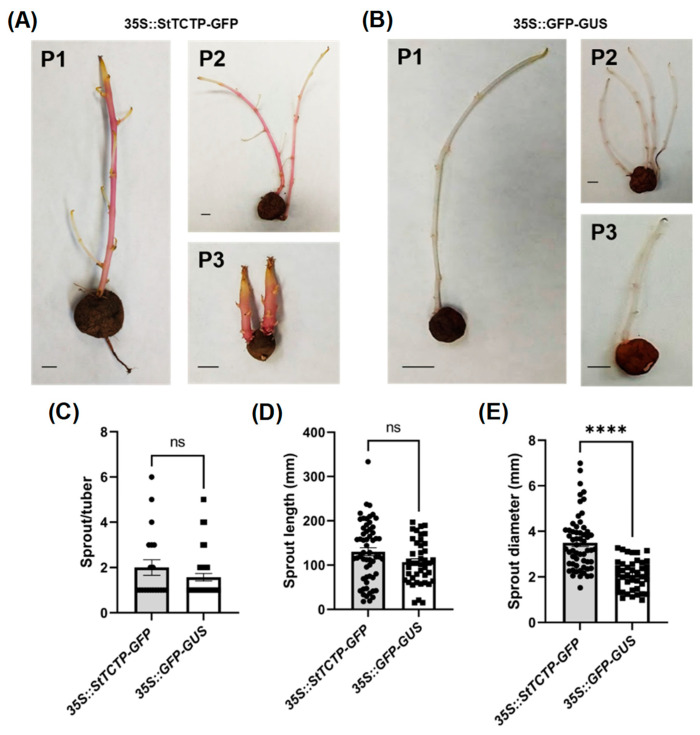
Phenotype of tubers from agroinfiltrated plants. Tubers were collected two months after agroinfiltration. These were incubated at 4 °C in dark conditions to visualize sprout emergence in plants (P1–P3) agroinfiltrated with *35S::StTCTP-GFP* (**A**) and *35S::GFP-GUS* (**B**). Number of stolons per tuber (**C**) and sprout length (**D**) and diameter (**E**) were determined in plants agroinfiltrated with either *35S::StTCTP-GFP* or *35S::GFP-GUS* constructs; data dispersion is represented with black circles and squares, respectively. Scale bars = 1 cm. Sprout/tuber (*35S::StTCTP-GFP* n = 20; *35S::GFP-GUS* n = 37). Sprout length and diameter (*35S::StTCTP-GFP* n = 58; *35S::GFP-GUS* n = 42). Bars represent standard error of the mean. Asterisks indicate significant difference according to Mann–Whitney tests: **** *p* < 0.0001; not significant (ns).

## Data Availability

The data will be available from authors on request.

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
