# Peer review of "Long-Distance Movement of Solanum tuberosum Translationally Controlled Tumor Protein (StTCTP) mRNA"

_plants, 2023, doi:10.3390/plants12152839_

Round 1

Reviewer 1 Report

The manuscript entitled “Long distance movement of Solanum tuberosum Translational Controlled Tumor Protein (StTCTP) mRNA” submitted for publication by Xoconostle-Moran and coworkers investigates the impact of the StTCTP gene, which is assumed to promote regeneneration processes, on potato tuber physiology.

Obviously, the authors are quite familiar with the family of TCTP proteins, but not experts in the research field of potato physiology.

The introduction focuses on the role of TCTPs in animals (why?), whereas the introduction in phloem-mobile tuber inducing signaling molecules is rather poor. The list of phloem-mobile tuber inducing signals is far from being complete (p. 2).

The authors are not aware of the fact that potato genome encodes as much as 9 different FT homologues and it would be important to mention which ones are important for tuberization.

The authors tried to demonstrate phloem-mobility of the StTCTP mRNA by RT-PCR of infiltrated leaves compared to apical leaves (called here “systemic leaves” for unknown reasons) and roots. However, the phloem-mobility of the mRNA is not shown convincingly. What is shown in Figure 2A? Is this quantification of the endogenous StTCTP manuscript? What is shown in 2C? What is encoded by the bar gene? Is this the BASTA resistance gene of the vector used for transformation? The experiments and control experiments are not explained sufficiently.

In the conclusion section it is mentioned that “StTCTP mRNA is transported long-distance to the apex” (p. 11), but the transport into the apex was not investigated.

The statement “StTCTP induces…..early stages of tuberization” is not comprehensive neither. The stages of tuberization strongly depends on the time point of harvest. If the amount of early stages tubers is higher, than the time point of tuber induction is delayed in this set of plants or tuber development is disturbed.

Minor points:

Nomenclature of transiently expressed constructs is misleading. StTCTP::GFP suggests usage of the StTCTP promoter for GFP expression, but here, the CaMV 35S promoter has been used. Therefore, the construct should be named “35S::StTCTP-GFP”. The same holds true for the GFP-GUS construct.

The term “seedling” in potato is also misleading and should be replaced by “sprout”.

mRNA-protein interactions are predicted in silico by molecular docking simulation using appropriate software tools, but experimental confirmation is missing.

The proteins PTB1 and PTB6 are mentioned on page 7 of the manuscript without any introduction or justification. The physiological function of PTBs are only explained on page 9 of the manuscript.

Therefore, the manuscript still needs thorough improvement in order to be suitable for publication.

Reviewer 2 Report

This study reported the movement of the TCTP transcript in potato and its effect on tuberization and root regeneration. The paper was very well written and could be interesting to the area of phloem signaling and plant development. However, I have these few questions listed below.

1) What is the length difference between GUS and TCTP? Is it possible that the immobility of GFP:GUS is due to its larger size (if it is larger than TCTP)?

2) How can you be sure that the underground phenotype was caused by the movement of TCTP mRNA rather than the protein? Detection of TCTP mRNA in the systemic tissues does not necessarily mean than the protein is not involved in the process. 

Reviewer 3 Report

I recommend the publication of the article because scientific experimentation on TCTP, which regulates cell cycle progression and programmed cell death and is involved in, among other functions, plant regeneration and the response to abiotic and biotic stresses, appears to be really interesting and topical. The aim and objectives of the article have been stated and are very fascinating. The use and study of the TCTP protein can be used for the genetic and metabolic study of many eukaryotic taxa. Furthermore, many molecular aspects have not yet been highlighted. The work is certainly of international interest and the format applied is certainly suitable for a research article. The work is original, of particular interest and can certainly stimulate research on this topic. The length of the article is good for the journal and the graphs and tables are clear and easy to understand. The conclusion summarises the aims of the work and future prospects.

Round 2

Reviewer 1 Report

The revised manuscript has been thoroughly improved.

Reviewer 2 Report

Dear authors,

Thanks for trying to address my questions. I know those two questions are not easy to answer, but I believe your way of handling them is honest and professional. At the current stage, I think it is great although more experiments will need to be done if a more definitive answer is pursued.